# The Foliar Application of Rice Phyllosphere Bacteria induces Drought-Stress Tolerance in *Oryza sativa* (L.)

**DOI:** 10.3390/plants10020387

**Published:** 2021-02-18

**Authors:** Arun Kumar Devarajan, Gomathy Muthukrishanan, Jaak Truu, Marika Truu, Ivika Ostonen, Subramanian Kizhaeral S., Periyasamy Panneerselvam, Sabarinathan Kuttalingam Gopalasubramanian

**Affiliations:** 1Department of Agricultural Microbiology, Tamil Nadu Agricultural University, Coimbatore 641003, India; arun.kumar.devarajan@ut.ee; 2Institute of Molecular and Cell Biology, University of Tartu, 51010 Tartu, Estonia; jaak.truu@ut.ee (J.T.); marika.truu@ut.ee (M.T.); 3Department of Soil Science and Agricultural Chemistry, Agricultural College and Research Institute, Tamil Nadu Agricultural University, Killikulam, Tuticorin 628252, India; gomathymicro@gmail.com; 4Institute of Ecology and Earth Sciences, University of Tartu, 51003 Tartu, Estonia; ivika.ostonen@ut.ee; 5Department of Nano Science and Technology, Tamil Nadu Agricultural University, Coimbatore 641003, India; kssubra2001@rediffmail.com; 6Crop Production Division, ICAR—National Rice Research Institute, Cuttack, Odisha 753006, India; panneerccri@rediffmail.com; 7Department of Plant Pathology, Agricultural College and Research Institute, Tamil Nadu Agricultural University, Killikulam, Tuticorin 628252, India

**Keywords:** plant-growth-promoting bacteria, phyllosphere, *Bacillus megaterium* PB50, drought stress, induced systemic tolerance

## Abstract

This study assessed the potential of *Bacillus endophyticus* PB3, *Bacillus altitudinis* PB46, and *Bacillus megaterium* PB50 to induce drought tolerance in a susceptible rice cultivar. The leaves of the potted rice plants subjected to physical drought stress for 10 days during the flowering stage were inoculated with single-strain suspensions. Control pots of irrigated and drought-stressed plants were included in the experiment for comparison. In all treatments, the plant stress-related physiochemical and biochemical changes were examined and the expression of six stress-responsive genes in rice leaves was evaluated. The colonization potential on the surface of the rice leaves and stomata of the most successful strain in terms of induced tolerance was confirmed in the gnotobiotic experiment. The plants sprayed with *B. megaterium* PB50 showed an elevated stress tolerance based on their higher relative water content and increased contents of total sugars, proteins, proline, phenolics, potassium, calcium, abscisic acid, and indole acetic acid, as well as a high expression of stress-related genes (*LEA*, *RAB16B*, *HSP70*, *SNAC1*, and *bZIP23*). Moreover, this strain improved yield parameters compared to other treatments and also confirmed its leaf surface colonization. Overall, this study indicates that the foliar application of *B. megaterium* PB50 can induce tolerance to drought stress in rice.

## 1. Introduction

Rice (*Oryza sativa* L.) is the most important food crop in the world and a staple food source for over half of the world’s population. It is estimated that by 2050, rice production will have to increase by 40% to meet the food requirements of 9 billion rice consumers [1]. Rice crops consume more water than other crops and in fact, 90% of rice is grown in Asia, which consumes approximately 80% of the world’s total irrigation water resources [2]. In addition to the growing world population, water shortages and climate change are creating more challenges for the agricultural system, representing barriers to solving food insecurity [3]. According to one report, in the near future 11% of current cropland, including regions in Africa, the Middle East, China, Europe, and Asia, is vulnerable to water scarcity [4]. Regarding abiotic stress, agricultural drought is a significant problem that results from an extended period of abnormally low rainfall or water shortage. This is the situation in which the rate of evapotranspiration is high and the rate of soil water consumption by crops is low [5]. To address this problem, new rice cultivation patterns must be developed to increase water use efficiency, new rice cultivars have to be identified, or existing rice cultivars have to be enhanced to ensure better crop yields under water deficit conditions.

Drought stress during the vegetative period of rice crop, particularly during the booting phase, interrupts flowering and results in spikelet sterility, poor grain filling, reduced grain weight, and, ultimately, poor yield [6]. For better adaptation and tolerance to drought stress, plants should undergo physiological, biochemical, and molecular changes, and one of the crucial factors influencing these changes is abscisic acid (ABA), which provides a vital function in the management of drought stress. ABA levels are abundant under water deficit conditions, stimulating the closure of stomata and inducing the expression of stress-related genes via transcription factors (TFs) [7]. The ABA-dependent signaling pathway is the most common mechanism in plants for overcoming drought stress however, plants also use ABA-independent signaling pathways to activate stress-related genes [8]. Important TFs involved in the regulation of drought stress are AREB, AP2/ERF, NAC, bZIP, MYC, and MYB, which are mediated by ABA-dependent and -independent pathways and often cross paths at different stages [9,10]. TFs further induce the expression of genes involved in stress-responsive mechanisms, such as antioxidants, molecular chaperones, ion channel proteins, and osmoprotectants, and help to develop stress tolerance in plants [11,12]. Understanding these mechanisms in detail can help in developing drought-tolerant rice plants using modern molecular biology techniques.

However, plant growth-promoting (PGP) microbes are gaining prominence in crop improvement and stress management, given the time and regulation constraints of plant breeding and genetic engineering technology [13]. Among PGP microbes, the beneficial effects of bacteria have been widely studied. A wealth of research has been conducted to advance the field of induced systemic tolerance against abiotic stress by rhizosphere bacteria [14,15,16,17]. Only limited research has been conducted in the area of phyllosphere microbiology. After first understanding the ability of phyllosphere microbes to alter the plant system and biogeography by influencing plant performance in various environmental circumstances, the phyllosphere microbiome began to receive considerably more attention [18]. The importance of endophytic phyllosphere microbes in promoting plant growth and relieving stress through seed inoculation has been suggested by many studies, and no studies have yet been conducted where microbial foliar application was used to induce systemic tolerance to drought stress in plants [19,20]. Most of the epiphytic phyllosphere bacterial applications studies were done with *Methylobacterium* sp. with other genera being concealed [21]. The role of epiphytic phyllosphere microbes in beneficial interactions and the underlying mechanisms are still poorly understood.

The flowering stage of rice is highly sensitive to environmental stress, particularly drought stress, which causes significant yield loss [22]. Therefore, it is necessary to study the influence of phyllosphere microbes on the physiological, biochemical, and molecular changes occurring in rice during flowering that allow adaptation to drought stress for potential application in mitigating yield loss. To the best of our knowledge, no detailed study has been conducted to identify the role of epiphytic phyllosphere bacteria in establishing induced systemic tolerance in rice. In our earlier study, we isolated efficient abiotic-stress-tolerant rice phyllosphere bacteria, and their potential to alleviate drought stress in rice was partially elucidated [23].

The objective of this study was to first investigate the effect of applying foliar to three epiphytic phyllosphere bacteria from genus *Bacillus* on stress-related physicochemical and biochemical modification, and gene expression in rice leaves under moderate physical drought stress during the flowering stage. Finally, the study’s objective is to also confirm the potential for a leaf surface colonization of one strain that effectively induces a stress-tolerance mechanism in rice.

## 2. Results

### 2.1. Plant Stress-Related Attributes of Rice Leaves under Different Treatments

The physicochemical and biochemical analyses revealed that the values of the examined parameters did not vary considerably between the replicates of each treatment, with significant differences observed in the rice leaves of different treatments after 10 days of drought stress (Table 1). The relative water content (RWC) was significantly higher in irrigated control plants (Ci) and the lowest RWC was recorded in drought-stressed control plants (Cws) (Table 1, Appendix A). Regarding the applications of different bacterial strains under drought stress, plants sprayed with *B. megaterium* PB50 had significantly higher RWC than plants sprayed with *B. endophyticus* PB3 and *B. altitudinis* PB46, reaching 2.4% and 5.5% respectively, whereas in *B. megaterium* PB50-treated plants, RWC was in a similar range as irrigated control plants. A significantly higher increase in potassium content compared to stress control plants (Cws) was recorded in plants treated with the PB50 strain compared to treatments with strains PB3 and PB46, reaching 19.2% and 7.6% on average, respectively (Table 1, Appendix A). The maximum calcium content was recorded in PB50-sprayed plants, whereas plants treated with strain PB3 and drought-stressed control plants (Cws) showed no significant difference in the leaf calcium concentrations (Table 1, Appendix A).

We found significant differences in the concentration of proteins between all treatments, especially in plants under drought stress with bacterial application (Table 1, Appendix A). The concentrations were highest in the plants treated with PB50 spray, which exceeded the stress control plants (Cws) by 26.6%. The leaves of the plants treated with the PB3 and PB46 strains had significantly higher (8.4% and 3.9%, respectively) protein concentrations than in the stress control plants (Table 1, Appendix A). Total soluble sugars (TSS) was also highest in the plants treated with PB50 followed by PB3-sprayed plants. Comparatively, drought-stressed control plants contained lower levels of protein and TSS than foliar-inoculated stressed plants.

The irrigated control plants (Ci) had the lowest proline concentrations (Table 1, Appendix A). The plants with the foliar application of PB50 had significantly higher proline contents than the those subjected to other treatments and was 25.0% higher than in the drought-stressed control plants (Cws). Interestingly, there was no significant difference in proline accumulation between PB3 and PB46 foliar applications. Conversely, the assessment of total phenolic compounds depicts a significant difference in the concentrations of phenolics between plants of all treatments (Table 1, Appendix A). The rice sprayed with PB50 had the highest total phenolic content, followed by PB3- and PB46-sprayed plants. The phenolic content in the Cws treatments was lower than in all plants receiving bacterial application.

The level of MDA, indicating the rate of lipid peroxidation due to oxidative damage during stress, was the highest in the drought-stressed control plants (Cws). The MDA content was significantly lower in the plants sprayed with PB3, PB46, and PB50 under drought stress (25.6%, 21.6%, and 33.8%, respectively) than in the drought-stressed control plants (Table 1, Appendix A). The irrigated control plants (Ci) and with bacterial application had the lowest MDA content. The lowest indole acetic acid (IAA) and ABA contents were recorded in plants with the Ci treatment; in the leaves of drought-stressed control plants in the Cws treatment, the contents were significantly higher (12.0% and 15.5%, respectively) (Appendix A). Under drought stress, the maximum values for IAA and ABA were recorded in PB50-treated plants, followed by PB3- and PB46-treated plants.

The measurements of enzyme activities show significant differences between treatments (Table 2, Appendix A). All three measured activities were the highest in the drought-stressed control plants and lowest in the controls without drought stress and bacterial application. From the treatments with bacterial application, significantly higher ascorbate peroxidase (APX) activity was recorded in the PB3-treated plants and significantly higher catalase (CAT) activity was recorded in the PB46-treated plants, while the glutathione peroxidase (GPX) activity in plants was not significantly different for these treatments.

PCA ordination based on the physicochemical and biochemical parameters resulted in the samples of rice leaves treated with strains PB3 and PB46 gathering into one cluster on the PCA plot (Figure 1). The PB50-treated samples clustered separately from the other treatments (especially from the control treatments) due to the higher values for physiochemical and biochemical parameters (separation along PCA axis 1) in leaves. The cluster of irrigated control plants was located separately from the other treatments (especially from the drought-stressed control plants) due to the higher RWC content of leaves (separation along the PCR axis 2), and the drought-stressed control plants were distinct from the other treatments (especially from the irrigated control plants) based on higher enzyme activities.

### 2.2. Expression of Stress-Related Genes

The expression of six stress-responsive genes in the third and fourth young leaves was assayed to verify the results of the biochemical analyses. The results show that all target genes were upregulated in all treatments. The relative upregulation differed between genes and was also significantly different between treatments (Figure 2, Appendix A). Drought-stressed plants treated with PB50 foliar spray showed the highest expression of almost all target genes. The exceptions were *AP2/ERF*, the expression of which wase significantly higher compared to the other treatments in the drought-stressed control plants (Cws) and lowest in the PB50-treated plants, and *SNAC1*, whose relative expression was highest in PB3-treated plants. Clustering of the samples according to the relative expression of a set of genes showed that the replicates of each treatment formed a distinctive cluster on the heatmap (Figure 3). A higher similarity in the relative expression of genes between the rice leaves of PB50 and PB3 treatments was indicated by the clustering of these samples, while the PB46 treatment showed greater similarity to the drought-stress controls.

Co-inertia analysis (CIA) was applied to assess the correspondence of two datasets (expression of the stress-related genes and plant chemical and biochemical data). The graphical output from this analysis is provided in Figure 4. The CIA results showed a high correlation (RV = 0.91, *p* < 0.001) of the two datasets. The first CIA axis captures the most variance (86.6%) in the two datasets and separates the PB50 treatment from the rest of the treatments (Figure 4E). The second CIA axis (13.1%) emphasizes the difference between drought-stressed control plants and other treatments. The CIA results indicated a high co-variation of most measured plant chemical and biochemical variables with expression of the stress-related genes, except for *AP2/ERF* (Figure 4C,D).

### 2.3. Yield Parameters

To understand the impact of treatments on crop resilience and improvement, yield parameters, such as panicle length and weight, and 100-grain weight, were measured on 97 days after sowing (DAS). Plants foliar-sprayed with PB50 had the highest yield parameters. The measured values for panicle length, panicle weight, and 100-grain weight following this treatment significantly exceeded the corresponding values for irrigated control plants by 4.9%, 22.6 %, and 17.2%, respectively, which presented the lowest values of all treatments (Figure 5). Although the panicle length and grain weight were comparable to those for the PB50- and PB3-treated plants, the latter showed significantly lower panicle weight than those treated with strain PB50. The yield parameters of the PB46-treated plants were significantly lower than those of the other two strain treatments, showing a similar range to the drought-stressed control plants.

### 2.4. Association of PB50 Strain on Rice Leaf Surface and Its Impact on Stomatal Closure

The impact of *B. megaterium* PB50 on the rice stomatal complex under normal and osmotic stress conditions was assessed in a gnotobiotic experiment. The results of this experiment show that the stomatal aperture size differed significantly between plants subject to different treatments; the smallest aperture was detected in plants treated with the spray containing strain PB50 (Figure 6). The size of the stomatal opening in plants sprayed with PB50 was 0.810 ± 0.173 μm under non-stressed condition (T_3_) and 0.348 ± 0.103 μm under osmotic stress (T_4_), which are 1.9- and 3.7-fold lower than in the normal (T_1_) and drought-stressed control plants (T_2_), respectively (Figure 7). The bacterial colonies were observed in the ridges of rice leaf (Figure 8A), on trichomes (Figure 8B), near silica bodies (Figure 8C), and near stomata produced with exopolysaccharides (EPS) (Figure 8D).

## 3. Discussion

### 3.1. Effect of Bacterial Application on Physicochemical and Biochemical Properties Related to Stress Tolerance in Rice Leaves

In plants, the effective retention of relative water content is a major challenge in achieving drought stress tolerance. The foliar application of plant growth-promoting bacteria has been suggested to improve plant growth and, thereby, crop yield [24,25,26]. We investigated the effect of the foliar application of three phyllosphere bacterial strains from the genus *Bacillus* on rice plants in terms of allowing them to overcome drought stress through the assessment of related parameters. The described properties of these strains allowed us to consider them as potential candidates for this challenge [27]. The applied set of analytical tools in this experimental study revealed that all three strains induced drought-tolerance mechanisms in rice plants. The RWC as well as the contents of chemical and biochemical indicator compounds were higher in almost every case for plants in all three treatments compared to the plants growing under drought stress. However, the highest positive effect was detected for *B. megaterium* PB50.

During drought stress, plants protect their cells from oxidative damage using several mechanisms. We found that the contents of potassium and calcium were increased in the PB50 sprayed rice plants compared to the plants of the other treatments and especially the control plants. Potassium is an inorganic osmolyte, and increases in potassium content in *Lavandula dentata* treated with *Bacillus thuringiensis* have been shown to relieve drought stress [28]. The authors suggested that the applied bacteria may produce enzymes to solubilize inorganic potassium in the soil and promote the potassium uptake of plants. Moreover, the sprayed bacteria can produce extracellular polymeric substances (EPS), which can store calcium in rice leaves by forming carbonate precipitation in the vicinity of bacterial cell walls [29].

In plants, sugars and proteins play a major protective role during abiotic stress by stabilizing the cell membrane and preventing oxidative damage [30,31,32]. The increase in protein and sugar contents after foliar application of plant-growth-promoting *Bacillus* during moisture stress has previously been documented in rice leaves [27,33]. The contents of sugars, proteins, prolines, and phenols were also highest in the PB50-sprayed rice plants. Similarly, an increase in RWC and the contents of proline, sugars, proteins, and amino acids after the inoculation of maize with plant-growth-promoting bacteria (PGPB) was shown to mitigate drought stress [34,35]. In this study, the content of total phenolic compounds was found to be higher in PB50-treated rice plants. Similarly, the foliar application of plant growth-promoting rhizobacteria (PGPR) was reported to increase total phenolics in pea plants and promote the plant’s resistance to powdery mildew [36]. The inoculation of PGPR on the roots of *Mentha piperita* increased the total phenolic content in plants and, thus, alleviated drought stress [37].

The malondialdehyde (MDA) content in plant leaves is an indicator of the level of lipid peroxidation during oxidative damage. Interestingly, the stress control plants had a significantly higher rate of lipid peroxidation compared to other treatments, while the irrigated control and PB50-sprayed plants had the lowest lipid peroxidation levels. Previous studies in rice plants have shown that seeds inoculated with PGPR and phyllosphere bacteria were more stress-tolerant and had reduced levels of salinity and, in particular, lipid peroxidation [27,38].

In general, antioxidant enzymes are synthesized to reduce the levels of reactive oxygen species in plants during stress. In this study, the activities of APX, CAT, and GPX enzymes in drought-stressed control plants significantly exceeded the activities in the plants with bacterial treatments, which synthesized more osmolytes instead of producing antioxidant enzymes and had a lower level of lipid peroxidation, suggesting reduced oxidative damage in foliar-sprayed plants, and that they experience less stress compared to the drought-stressed control plants. A similar conclusion was reported by Saikia et al. [39] and Naseem and Bano [35].

In our study, the abscisic acid (ABA) content was prominent in PB50-treated plants under drought stress conditions. ABA is a plant hormone that plays a major role in stomatal movement and provides adaption to abiotic stress. The ability of *B. megaterium* species to influence the plant endogenous ABA content was previously reported by Porcel et al. [40]. ABA-producing bacteria, such as *Azospirillum brasilense*, were shown to mitigate drought stress in *Arabidopsis thaliana* by enhancing plant ABA content [41]. In this study, the production of IAA was found to be higher in bacteria foliar-sprayed plants than in control treatments. Under stress conditions, the increased indole acetic acid (IAA) induces the production of 1-aminocyclopropane-1-carboxylate-diaminase (ACC-deaminase) in the bacterial cell, and the ACC is shunted from ethylene, an inhibitor of plant growth, to ammonium and α-ketobutyrate production [42]. A report on the exogenous application of ABA and sucrose showed increased starch synthesis in the panicle and improved grain and yield quality in rice [43]. In the current study, the higher contents of ABA and total sugars in rice plants sprayed with phyllosphere bacteria, especially with strain PB50, probably contributed to the increase in yield parameters.

### 3.2. Effect of Bacterial Application on the Expression of Stress-Related Genes in Rice Leaves

All six genes assessed in the expression studies were found to be highly expressed in the leaves of rice growing under drought stress. All of these genes are known to play a crucial role in managing drought stress. *OsbZIP23* encodes a basic leucine zipper transcription factor in rice that is also known as an abscisic-acid-binding factor (ABF) [44]. The ABF transcription factor has been shown to induce abiotic stress tolerance through chaperone and osmolyte synthesis via the ABA-dependent pathway [45,46]. In our case, expression of the *bZIP23* gene was upregulated in plants of all treatments however, the expression was remarkably higher in PB50-sprayed plants, indicating the significant role of PB50 in alleviating drought stress in rice. Another study investigating rice seeds treated with *Pseudomonas fluorescens* pf1 reported the highest expression of the *bZIP* gene [47]. During drought, excessive ethylene induces premature leaf senescence and reduces crop yield. Drought also increases the expression of the gene encoding the ethylene response factor (*ERF*). However, as discussed above, the ACC-deaminase-producing bacteria can break down the ethylene precursor molecule ACC and, thus, reduce *ERF* expression [48]. Interestingly, a substantially lower *AP2/ERF* expression was observed in plants sprayed with the PB50 strain compared with the drought-stressed control plants, with this gene being the most highly expressed of the target genes assayed in this study. Vaishnav and Choudhary [49] reported higher ethylene production and lower expression of the gene encoding ethylene-responsive element-binding factor (EREB) in soybean inoculated with *Pseudomonas simiae* compared with uninoculated plants under drought stress. Based on the CIA results (Figure 4), *AP2/ERF* was the only gene of those studied whose expression did not co-vary with most measured plant chemical and biochemical variables. Plant-specific NAC transcription factors are encoded by *NAC* genes and regulated by the ABA-dependent and -independent pathways responsible for activating abiotic stress-responsive genes in rice plants [50]. In our study, the high expression of the SNAC1 gene in PB3- and PB50-treated plants could be related to an activation of the ABA-dependent pathway as the ABA content was high in these plants. In previous reports, an increase in the expression level of NAC TFs was documented during the interaction of plants with viruses, fungi, and bacteria [51]. The up regulation of NAC protein genes in bentgrass by synthetic PGPR volatile compound application has been reported [52]. Through the release of volatile compounds, the PB50 strain may have triggered SNAC1 genes.

*LEA* group genes are regulated by the ABA hormone [53,54]. We found that *LEA* and *RAB16B* genes were highly expressed in all treatment plants. PB50-sprayed plants also had high ABA content and showed the highest expression of these genes. Similarly, rice seedlings treated with *Bacillus amyloliquefaciens* and grown under osmotic stress exhibited induced tolerance through an induction of stress-responsive genes, such as *DHN* and *LEA* [55]. HSP70 proteins play a major role in the correction of denatured and misfolded proteins during environmental stress in living organisms [56]. In our case, the expression of the *HSP70* gene was the highest in PB50-treated plants under drought stress. The application of *P. fluorescens* on banana during thermal shock and rice during drought stress resulted in the significant upregulation of *HSP70* and *HSP20* genes, respectively [46,57]. In many crops, including rice, an improvement in yield through the application of plant growth-promoting bacteria was achieved [58].

### 3.3. Effect of Bacterial Application on Stomatal Closure

The findings of this study indicate that *B. megaterium* PB50 had a profound effect on rice stomatal closure for avoiding water loss due to evapotranspiration during drought-induced stress. Phyllosphere microbes have previously been shown to induce stomatal closure via an ABA-dependent pathway [59,60], and the results of our study suggest this is the case with *B. megaterium* PB50. The bacterial cells were found on the stomatal complex, leaf surface ridges, silica bodies, and trichomes. The role of *Bacillus amyloliquefaciens*-produced acetoin in inducing the stomatal closure of *Arabidopsis thaliana* and *Nicotiana benthamiana* was reported by Wu et al. [61]. Similarly, the PB50 strain was also reported to efficiently produce 3-hydroxybutanone (acetoin), under osmotic stress [23]. These regions are known to be the nutrient leakage areas, and plants usually hold water in trichomes during drought stress [62,63]. The diffusion of nutrients along with water droplets and volatiles on the surface of the leaf should create a favorable habitat for phyllosphere bacteria. However, rice plants release numerous volatiles, including methanol, through leaf surfaces [64,65] that can be used as a carbon source by PB50 [66]. The SEM colonization study suggested that EPS-producing *B. megaterium* PB50 could use volatiles from rice plants and colonize the leaf surface through biofilm formation on rice leaves. Altogether, we have also predicted the possible stress-tolerance inducing mechanism by the PB50 strain in rice with our findings (Figure 9).

## 4. Materials and Methods

### 4.1. Experimental Design and Preparation of Bacterial Inocula

Induced systemic tolerance (IST) in rice through the foliar application of phyllosphere bacteria strains during the flowering stage under moderate drought stress conditions was studied in a 97-day-long greenhouse pot experiment at the Agricultural Center and Research Station (8°42′ N, 77°51′ E), Killikulam, Thoothukudi, India, in 2019. The high-yielding popular rice variety CO51 had no desired drought tolerance (seeds were purchased from the Department of Plant Breeding and Genetics, AC & RI, Killikulam, Thoothukudi, India). The bacterial strains *Bacillus endophyticus* PB3, *Bacillus altitudinis* PB46, and *Bacillus megaterium* PB50 (MK969113, MK979282, and MK979284, respectively), previously isolated from rice leaves of drought-tolerant varieties and shown to have the potential to tolerate abiotic stress and possess diverse plant growth-promoting traits [23], were used in this experiment.

For the experiment, uniform, healthy, and surface sterilized rice seeds were sown in 20 pots (16 cm × 26 × 10 cm) containing 10 kg of sieved and sterilized clay loam soil with a basal application of 0.8 g of urea, 0.5 g of phosphorous pentoxide, and 1.1 g of potassium oxide. The seeds were sterilized by immersing the seeds in 0.5% sodium hypochlorite solution for 10 min and then washing them well with sterile water followed by shade-drying using sterile tissue paper. Six seeds were sown into each pot.

The experiment comprised three different treatments with foliar inoculation of three different bacterial strains and two different controls (pots with drought stress without inoculation and irrigated pots without inoculation). All treatments and controls were performed in four replicates.

The rice plants reached their flowering stage 60 days after sowing (DAS). After this point, moderate stress was imposed on the pots with drought stress treatments for 10 days by maintaining the water holding capacity at 55–60% (plant water potential of −1.20 to −1.40 MPa). The amount of water used for irrigation to maintain the moderate drought stress was calculated applying the following formula [67]:Water for irrigation = D × H × A × (FCI − FC0)(1)
where D is the soil bulk density, H is the soil depth, A is the area of each pot, FC1 is the desired soil field capacity, and FC0 is the actual soil field capacity before irrigation.

The foliar inoculation of the bacterial strains was performed on the first and seventh day after drought stress initiation.

Three inocula for foliar application were prepared by growing the isolated bacterial strains in Tryptic soy broth (TSB) for 24 h. Then, the cells were pelletized by centrifugation (Remi CM-12 plus, Chennai India) for 10 min at 5000 rpm and then dissolved and suspended in sterile phosphate-buffered saline (PBS, 0.2 g L^−1^ NaCl, 1.44 g L^−1^ Na_2_HPO_4_, and 0.24 g L^−1^ KH_2_PO_4_). By using the growth data of bacterial strains (OD_600_), the number of cells in the suspension (colony forming units (CFU) mL^−1^) was measured as described by Sethuraman and Balasubramanian (2010) [68]. For foliar inoculation, a suspension with a concentration of approximately 10^9^ CFU mL^–1^ was prepared.

A low-volume sprayer was used to spray 15–25 mL of inoculant per plant. A similar amount of sterile PBS solution was sprayed on the control plants [69,70]. The treatments are designated in the text as follows: PB3—drought stress with *Bacillus endophyticus* PB3 foliar spray, PB46—drought stress with *Bacillus altitudinis* PB46 foliar spray, PB50—drought stress with *Bacillus megaterium* PB50 foliar application, Cws—drought stress control, and Ci—irrigated control.

After re-watering, the plants were allowed to grow until the harvest phase to study the impact of bacterial application and 10 days of drought stress on the yield of rice. Plants reached the harvest stage 97 DAS and, on day 97, the length and weight of the panicle, and 100-grain weight were recorded for each plant.

### 4.2. Sampling and Physiochemical Analysis of Leaves

After the 10th day of growth under drought stress conditions (70 DAS) and before re-watering, leaf samples (leaves without dryness) were collected for physiochemical, biochemical, and stress-responsive gene expression analyses of each treatment (Figure 10) and snap-frozen in liquid nitrogen and stored at −80 °C until further analysis.

The relative water content (RWC) was estimated on 70 DAS using fresh leaf samples. A total of 12 individual leaves from each treatment replicates were collected for this purpose and weighed to obtain fresh weight (FW). To measure the soaked weight (SW), the leaves were placed in distilled water for 16 h and re-weighed. After drying at 80 °C for 48 h in a hot air oven, the dry weight (DW) of the leaf was determined. The RWC was calculated using the formula given by Turner and soil (1981) [71], and the results are presented as percentages.

For the estimation of potassium (K) and calcium (Ca) contents in rice leaves, 1 g of dried leaf sample was digested in 12 mL of a perchloric acid, sulfuric acid, and nitric acid (1:2:9) mixture for 3 h in a cold bath and 3 h in a sand bath until complete digestion [72]. The concentration (ppm g^−1^ DW) of K was measured after the neutralization of digestate with ammonium hydroxide using a flame photometer equipped with a K lamp at 766 nm [73]. The Ca concentration (%) was estimated using the titration method proposed by Tucker and Kurtz (1961) [74].

### 4.3. Biochemical Analysis

#### 4.3.1. Determination of Total Protein, Total Soluble Sugar, and Proline Contents

The total protein content (µg g^−1^ FW) was measured at 650 nm against different concentrations of bovine serum albumen (BSA) using the standard Lowry et al. [75] procedure. The total soluble sugar content (µg g^−1^ FW) of rice leaves was measured at 420 nm with glucose as a standard using the phenol–sulfuric acid method described by Dubois. et al. [76]. Proline content in rice leaves was measured at 520 nm using acid ninhydrin and glacial acetic acid as described by Bates et al. [77]. Using the standard curve of free proline, the concentration of proline in the leaf samples was calculated and is expressed as µg g^−1^ FW.

#### 4.3.2. Total Phenolic Content

The Folin–Ciocalteu colorimetric approach was used to assess the content of total phenolics in rice leaves [78]. Briefly, the fresh leaves were ground with 80% ethanol and incubated for 2 h at 4 °C in tightly closed centrifuge tubes in the dark. The filtered leaf extract was then incubated in the presence of Folin–Ciocalteu reagent and 7.5% (wt/vol) Na_2_CO_3_ at 45 °C for 15 min. The absorbance of phenolic content was measured at 750 nm using gallic acid as the standard and the results are expressed as gallic acid equivalents per g of fresh leaf weight (mM GAE g^−1^ FW).

#### 4.3.3. Determination of Lipid Peroxidation

Lipid peroxidation was assessed by measuring the malondialdehyde (MDA) content in rice leaves [79]. For this, 100 mg of rice leaves from each treatment was homogenized with 0.5 mL of 0.1% (w/v) trichloroacetic acid (TCA) and centrifuged at 10,000 rpm for 10 min at room temperature. Then, the supernatant was mixed with 1.5 mL of 0.5% (w/v) thiobarbituric acid (TBA) and the sample mixtures were incubated in a water bath at 95 °C for 25 min. The reaction was terminated by incubating the samples on an ice bath, and the sample solutions were again centrifuged at 10,000 rpm for 5 min. The fluorescence intensity at 532 and 600 nm was measured in the supernatant using a spectrophotometer (LAMBDA 365 UV–Vis spectrophotometer, PerkinElmer, Mumbai, India). The MDA content was calculated as follows:(2)$$\mathrm{MDA}\;\mathrm{equivalent}\;\mathrm{content}\;\mathrm{in}\;\mathrm{leaf}\;\mathrm{tissues}=\frac{\left(Absorbance\;at\;523\;\mathrm{nm}-Absorbance\;at\;600\;\mathrm{nm}\right)}{155}\;\times \;1000.$$

The results are expressed as nmole of MDA equivalent per gram of fresh weight (nmol MDA mL^−1^ g FW^−1^).

#### 4.3.4. Estimation of IAA and ABA Hormones

Phytohormones were extracted and purified from the rice samples using the method described by Pan et al. [80]. From each treatment, 50 mg of leaf sample was taken and ground with liquid nitrogen. Then, an extraction solvent composed of propanol, H_2_O, and HCl (2:1:0.002) was added and incubated in a shaking ice bath for 30 min. Dichloromethane was then added to the solutions, which were again incubated on an ice bath for 30 min under shaking. Then, the solutions were centrifuged at 12,000 rpm for 5 min and the liquid layer containing phytohormones was separated and dried using a rotary film evaporator (RFE). Finally, the dry residue was resuspended in methanol and filtered through a Millipore filter (0.45 µm). The analysis was performed using ultra-performance liquid chromatography coupled with evaporative light scattering detection (UPLC–ELSD) with a C18 reverse-phase column (Shimadzu, Kyoto, Japan). Mobile phase A was distilled water with 0.1% (vol/vol) formic acid and mobile phase B was methanol with 0.1% (vol/vol) formic acid. IAA and ABA were extracted at 280 and 265 nm, respectively, and eluted at a flow rate of 0.8 mL min^−1^. The results are presented as µg mL^−1^.

#### 4.3.5. Assay of Antioxidant Enzymes

For the extraction of antioxidant enzymes, 1 g of leaf sample was collected and crushed in liquid nitrogen. The powdered leaf sample was mixed with 10 mL of phosphate-buffered saline (pH 7.0) and centrifuged at 10,000 rpm for 15 min at 4 °C. The supernatant was used as the enzyme extract in further measurements of enzyme activities [81]. The extract was stored at −80 °C. The total protein content in the enzyme extract was estimated using the method proposed by Bradford [82]. All the following activity measurements were based on colorimetry and measured using a spectrophotometer.

For ascorbate peroxidase activity (APX), 0.5 mL of enzyme extract was added to 5 mL of reaction mixture containing 0.2 mM of EDTA, 0.5 mM of ascorbic acid, 0.25 mM of H_2_O_2_, and 0.05 M of potassium phosphate buffer (pH 7.0) at room temperature. The reaction was started, and the absorbance was monitored spectrophotometrically at 240 nm for 1 min, as it is completely based on the reduction in the amount of ascorbate. An extinction coefficient of 2.8 mM cm^−1^ was used to calculate APX activity [83].

The catalase activity (CAT) activity was determined by monitoring the disappearance of H_2_O_2_ at 240 nm using the method defined by Aebi [84]. The reaction mixture consisted of 50 mM of phosphate buffer (pH 7.0), 33 mM of H_2_O_2_, and enzyme extract. The absorbance was recorded once the reaction began. An extinction coefficient of 36 M cm^−1^ was used to calculate CAT activity.

The guaiacol peroxidase activity was measured spectrophotometrically following Zaharieva et al. [85]. The reaction started by adding the enzyme extract to the reaction mixture consisting of 2.7 mM of guaiacol, 2 mM of H_2_O_2_, and 0.05 M of potassium phosphate buffer (pH 7.0) at room temperature. The absorbance was monitored spectrophotometrically at 470 nm for 1 min as the assay principle is based on the H_2_O_2_-mediated inhibition of guaiacol oxidation into tetraguaiacol. An extinction coefficient of 6.22 mM cm^−1^ was used to calculate GPX activity. Overall, all the three enzyme activities are expressed as variation in mmol per time unit per mg total protein (mmol min^−1^ mg protein^−1^).

### 4.4. RNA Extraction from Rice Leaves and Reverse Transcriptase Quantitative PCR (qPCR) Analysis

To estimate the expression of drought-stress-related genes in differently treated rice leaves, two-step qPCR was conducted with three replicates for each treatment. Total RNA was isolated from leaves of 70-day-old rice plants using the Nucleo-pore^TM^ RNASure Mini Kit (Genetix Biotech Asia Pvt. Ltd., Delhi, India) and following the standard kit protocol. The concentration of RNA was measured at 260 nm using a spectrophotometer (NanoDrop™ 2000/c Spectrophotometer, Thermo Scientific, Marietta, OH, USA).

The first strand of cDNA was synthesized using 1 μg of DNase-free total RNA primed with oligo dT primers in a 20 μL reaction mixture using RevertAid M-MuLV reverse transcriptase (RevertAid First strand cDNA synthesis kit, Thermo Scientific™, Marietta, OH, USA) and following the manufacturer’s instructions. QPCR was performed using Maxima SYBR Green/ROX qPCR Master Mix (Thermo Scientific™, Marietta, OH, USA) on instrument CFX96 Touch^TM^ (Bio-Rad, Des Plaines, IL, USA). The constitutively expressed *Actin* gene from the rice was used as the internal control. The primers targeting rice late embryogenesis abundant proteins (*LEA*), dehydrin Rab16B (*RAB16B*), 70 kilodalton heat shock proteins (*HSP70*), basic leucine zipper 23 (*bZIP23*), apetala2/ethylene-responsive factor (*AP2/ERF*), and stress-responsive NAC1 (*SNAC1*) genes were designed based on the respective gene sequences downloaded from the National Centre for Biotechnology Information (NCBI) using Primer-Blast (Table 3). The qPCR cycling conditions were similar for all genes and were as follows: Initial denaturation at 95 °C for 10 min; 95 °C for 30 s, and 60 °C for 1 min for 40 cycles; followed by melt curve analysis at 95 °C for 1 min, 60 °C for 30 s, and 95 °C for 30 s. All reactions were performed in three technical replicates. The transcript amount obtained for each target gene normalized to the internal control was examined using the 2^−ΔΔCT^ method [86] and the results are presented as relative expression fold-change.

### 4.5. Yield Parameters

After re-watering, the plants were allowed to grow until the harvest phase to study the impact of 10 days of drought stress on the rice yield. Plants reached the harvest stage at 97 DAS; on this day, the length and weight of the panicle and 100-grain weight were recorded for each treatment plant.

### 4.6. The Gnotobiotic Experiment for the Study of Bacteria Colonization on the Rice Leaf and Its Role of Stomatal Modulation

The leaf surface colonization by *B. megaterium* PB50 under normal osmotic stress was examined using scanning electron microscopy (SEM) of leaf tissue. Seeds of rice (CO51) were surface-sterilized using sodium hypochlorite. Seeds were sown in 16 phytoagar bottles (three seeds in each bottle) containing half Murashige and Skoog (MS) medium with 1% phytoagar in a 500 mL glass bottle (9.75 × 13.5 × 5.5 inches). Once the seeds germinated and attained the three-to-four leaf stage (15 DAS), the seedlings were transferred to another fresh MS phytoagar bottle with treatment conditions. The bacterial suspension was prepared in a mini spray bottle (50 mL) with 10^9^ CFU mL^−1^. A bacterial suspension spray (200 μL) was performed three times for each glass bottle and a phosphate-buffered saline solution (PBS) was sprayed for control treatment. The leaf harvesting was done after 24 hours of spray inoculation.

The treatments prepared in three replicates were as follows: T_1_—PBS spray; T_2_—PEG6000 (32.6%) + PBS spray; T_3_—PB50 spray inoculation; and T_4_—PEG6000 (32.6%) + PB50 spray inoculation. After 24 h of osmotic stress and PB50 spray inoculation, the bottles were taken for examination of the leaf samples by SEM.

The leaf samples were harvested and immediately fixed with 2.5% glutaraldehyde at 4 °C overnight, followed by dehydration of the leaf sample by washing serially with different ethanol concentrations of 30%, 50%, 70%, 90%, and 100% [87]. Finally, the samples were treated with 100% isoamyl acetate and dried. Then, the leaf samples were mounted with colloidal graphite placed onto an aluminum stub clamped on the transfer device. A thin layer of gold was applied over the samples using a SC7620 Mini Sputter Coater (Quorum Technologies Ltd., Lewes, United Kingdom). The sputter-coated leaf samples were examined under QuantaTM 250 FEG ESEM with Energy Dispersive Spectroscopy (EDAX) (FEI Company, Hillsboro, OR, USA) at an accelerating voltage of 10–20 kV [88].

### 4.7. Statistical Analysis

All the experiments were conducted with a minimum of four replicates for all experiments, and the results were expressed as the mean ± standard deviation (SD). Statistical analyses used were the one-way analysis of variance (ANOVA) followed by Duncan’s test, principal component analysis, co-inertia analysis, and multivariate analysis of variance (MANOVA) followed by multivariate pairwise comparison using R programming packages ade4 [89] and MANOVA.RM [90].

## 5. Conclusions

We found that the foliar application of *B. megaterium* PB50 produced the strongest effect in inducing drought stress tolerance in rice as evidenced by the enhancement of plant stress-tolerance-related parameters compared with uninoculated drought-stressed control plants. High levels of stress-tolerant-related physicochemical and biochemical parameters, except for MDA and antioxidant enzyme activity, were demonstrated by PB50-sprayed plants under drought conditions, indicating less oxidative damage caused by drought stress as osmolytes protected cells from damage. Our gene expression study showed that plants sprayed with the PB50 strain demonstrated the highest upregulation of stress-responsive genes, except for *AP2/ERF*, compared to control drought-stressed plants. Increased yield parameters for PB50-treated plants after restoration from drought stress indicated that PB50 not only protected plants from drought stress but also improved yield. Overall, *B. megaterium* PB50 colonization on the leaf surface and its potential alteration of the stomata and stress-tolerance-related parameters provide evidence for induced drought stress tolerance in rice. We conclude that the PB50 strain can serve as a useful foliar inoculant for sustainable agriculture in drought-prone regions where there are differences in rainfall and water supply.

## Figures and Tables

**Figure 1 plants-10-00387-f001:**
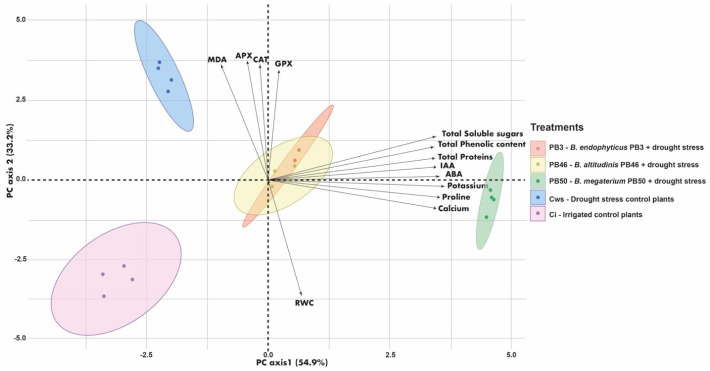
Plot of principal component analysis (PCA) showing ordination of the samples based on the physiochemical and biochemical properties of the plant leaves. Treatment are indicated by 95% confidence ellipses. RWC, relative water content; APX, ascorbate peroxidase activity; CAT, catalase activity; GPX, glutathione peroxidase activity; MDA, malondialdehyde; IAA, indole acetic acid; ABA, abscisic acid.

**Figure 2 plants-10-00387-f002:**
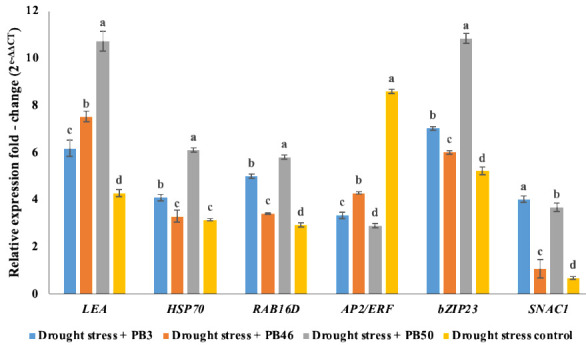
Bar graph of the relative gene expression of *LEA*, *HSP70*, *RAB16B*, *AP2/ERF*, *bZIP23*, and *SNAC1* in 70-day-old rice plants under different treatment conditions. Shown are means and standard deviation values (n = 4). Values with different letters are significantly different according to Duncan’s test (*p* ≤ 0.05).

**Figure 3 plants-10-00387-f003:**
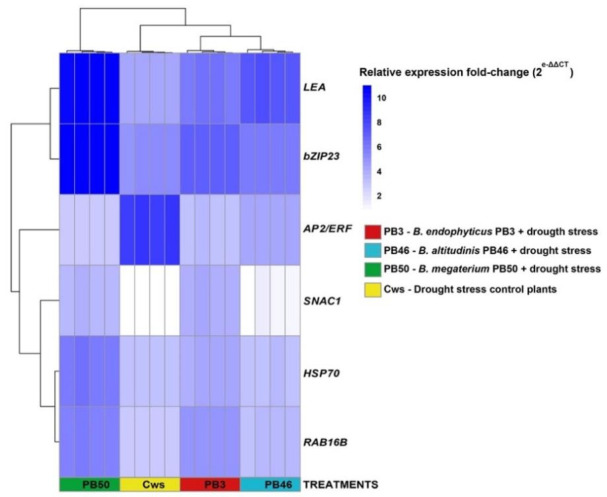
Heatmap showing the fold-change expression of genes in different treatments and clustering of samples according to the relative expression of the set of analyzed genes.

**Figure 4 plants-10-00387-f004:**
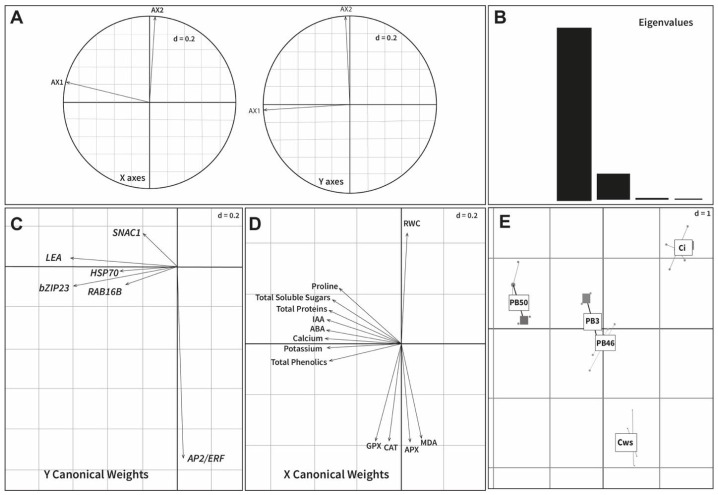
Co-inertia analysis (CIA) results based on two datasets (plant physico- and biochemical variables and gene expression data). (**A**) Projections of the principal axes of the two datasets onto the axes of the co-inertia analysis. X axes: Plant physico- and biochemical variables; Y axes: Drought-responsive gene expression data. (**B**) Scree plot of eigenvalues. (**C**) Correlation of gene expression data with the first two axes of the co-inertia analysis. (**D**) Correlation of plant physico- and biochemical variables with the first two axes of the co-inertia analysis. (**E**) Plot of the first two components in the sample space. Each sample is represented by a square, where the two datasets for each sample are connected by lines to a center point (global score).

**Figure 5 plants-10-00387-f005:**
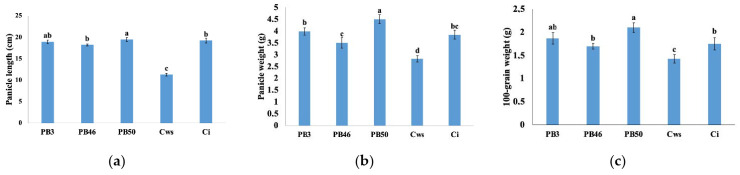
Bar graphs representing means and SD of the yield parameters: Panicle length (**a**), weight (**b**), and grain weight (**c**) of the rice plants of different treatments. PB3, drought stress with *B. endophyticus* PB3 foliar spray; PB46, drought stress with *B. altitudinis* PB46 foliar spray; PB50, drought stress with *B. megaterium* PB50 foliar spray; Cws, drought stress control; and CI, irrigated control. Values with different letters (above the bars) are significantly different according to Duncan’s test (*p* ≤ 0.05).

**Figure 6 plants-10-00387-f006:**
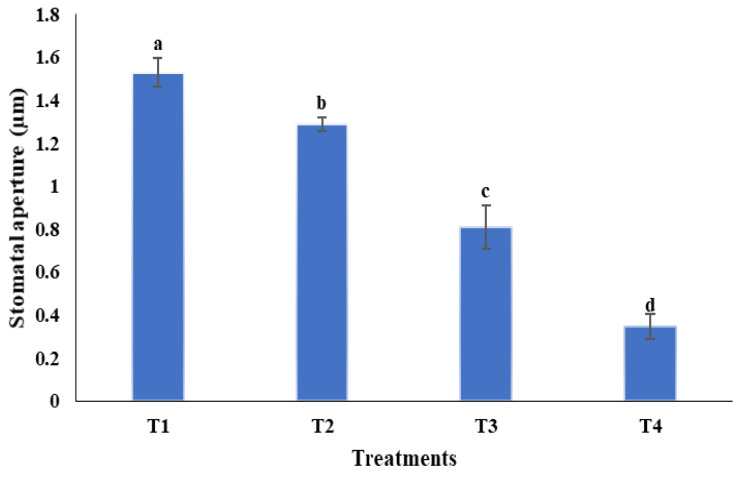
Bar graph showing means and SD of the stomatal aperture of the rice leaves of treatments under gnotobiotic conditions. T_1_—PBS spray; T_2_—PEG6000 (32.6%) + PBS spray; T_3_—PB50 spray inoculation; and T_4_—PEG6000 (32.6%) + PB50 spray inoculation. PBS, phosphate buffer solution. Values with different letters are significantly different according to Duncan’s test (*p* ≤ 0.05).

**Figure 7 plants-10-00387-f007:**
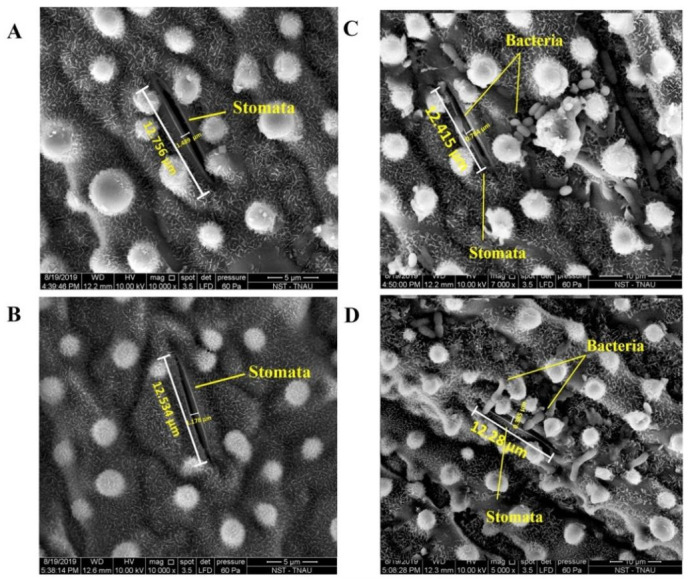
Representative SEM images showing the change in stomatal aperture in control rice plants under normal (**A**) and osmotic stress (**B**) conditions, and *B. megaterium* PB50 spray inoculated rice plants under normal (**C**) and osmotic stress (**D**) conditions.

**Figure 8 plants-10-00387-f008:**
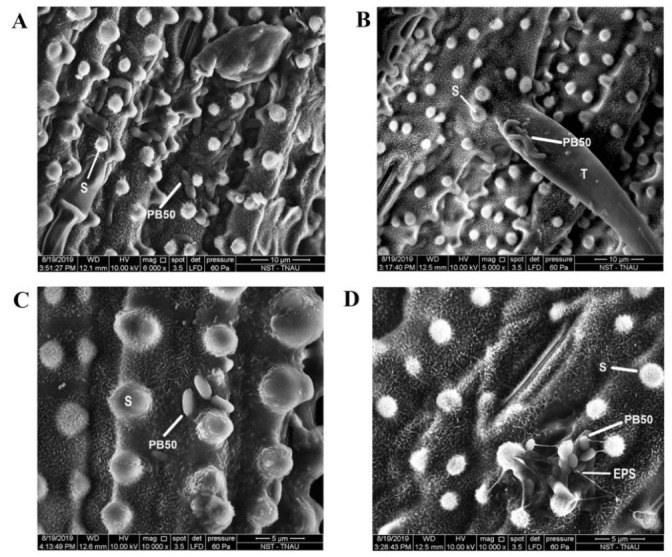
SEM images of *Bacillus megaterium* PB50 cells on the leaf surface of rice plants (T_4_). PB50 strain colonization in the ridges of rice leaf surface (**A**), on trichomes (**B**), near silica bodies (**C**), and near stomata produced with exopolysaccharides (EPS) (**D**). S, silica body; T, trichome; and EPS, exopolysaccharides.

**Figure 9 plants-10-00387-f009:**
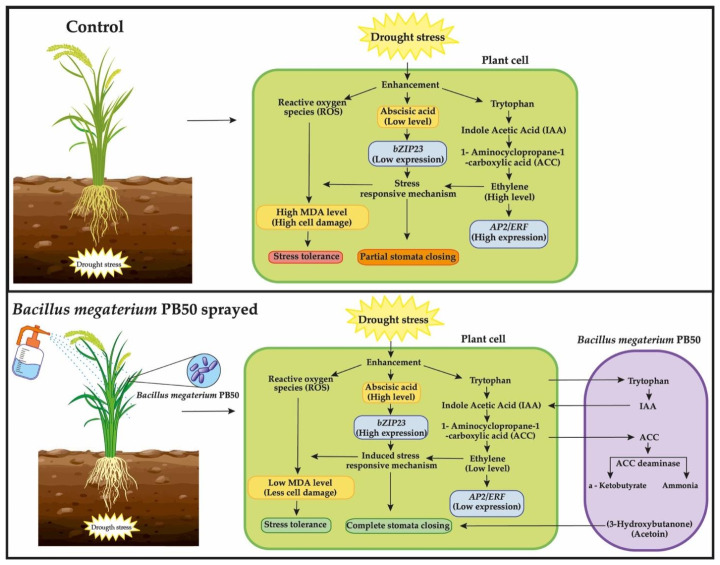
The figure presents schemes of two different mechanisms of rice plants under moderate drought stress conditions (−1.20 to −1.40 MPa) with and without foliar application of *B. megaterium* PB50 during the flowering period of rice plants (variety CO51). In control stress plants where the expression of the *bZIP23* gene is low due to the low inheritance potential to ABA production, excessive ethylene can be produced that induces the expression of *AP2/ERF* genes and causes leaf senescence. This stress-responsive mechanism is insufficient to protect plant cells, resulting in the excessive release of MDA from damaged cells due to reactive oxygen species. In the case of rice plants with leaves treated with a suspension of *Bacillus megaterium* PB50, the scenario can be just the opposite. The release of 1-Aminocyclopropane-1-carboxylate (ACC) deaminase by the PB50 strain reduces the expression of the *AP2/ERF* gene by breaking down the ethylene precursor molecule, and the other exogenous PB50 strain metabolites could induce ABA-dependent mediated stress-responsive mechanisms to reduce cell damage and induce stomatal closure to avoid evapotranspiration during drought stress.

**Figure 10 plants-10-00387-f010:**
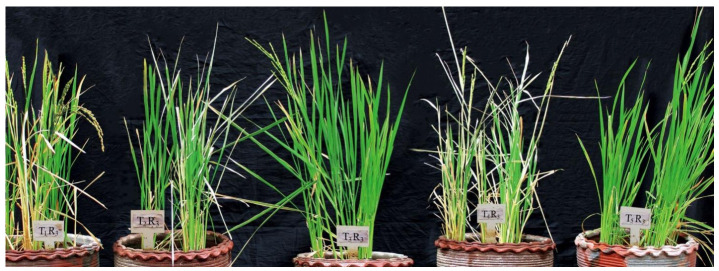
Photo of illustrating experimental setup. The rice plants during the flowering stage (70 days after sowing (DAS)) after 10 days of drought stress are shown. From left to right, the treatments are as follows: PB3, drought stress with *B. endophyticus* PB3 foliar spray; PB46, drought stress with *B. altitudinis* PB46 foliar spray; PB50, drought stress with *B. megaterium* PB50 foliar spray; Cws, drought stress control; and CI, irrigated control. R_3_ is the replicate 3 of each treatment.

**Table 1 plants-10-00387-t001:** Means and ± SD (n = 4) of the physicochemical parameters (RWC (relative water content), potassium, and calcium content) and contents of biochemical compounds (total proteins, total soluble sugars (TSS), total phenolics, prolines, malondialdehyde (MDA), indole acetic acid (IAA), and abscisic acid (ABA)), in the rice leaves of different treatments after 10 days of drought stress. PB3, drought stress with *B. endophyticus* PB3 foliar spray; PB46, drought stress with *B. altitudinis* PB46 foliar spray; PB50, drought stress with *B. megaterium* PB50 foliar spray; Cws, drought stress control and Ci, irrigated control; FW, fresh weight; GAE, gallic acid equivalent; DW, dry weight. Parameter values with different letters are significantly different according to Duncan’s test, *p* < 0.05.

Treatments	Physico-Chemical Parameters			Contents of Biochemical Compounds						
	RWC (%)	Potassium (ppm g^–1^ DW)	Calcium (%)	Total Proteins (µg g^–1^ FW)	TSS (µg g^–1^ W)	Prolines (µg g^–1^ FW)	Total Phenolics (mg GAE g^–1^)	MDA (nmol g^–1^ FW)	IAA (ng g^–1^)	ABA (ng g^–1^)
PB3	72.8 ± 0.6 ^c^	28.1 ± 0.1 ^c^	0.78 ± 0.01 ^bc^	277 ± 3 ^b^	2.99 ± 0.03 ^b^	28.3 ± 0.3 ^b^	2.57 ± 0.12 ^c^	3.04 ± 0.2 ^bc^	115 ± 4 ^b^	96 ± 2 ^b^
PB46	75.0 ± 1.1 ^bc^	31.1 ± 0.1 ^b^	0.84 ± 0.03 ^ab^	266 ± 4 ^c^	2.84 ± 0.02 ^c^	28.2 ± 0.4 ^b^	2.79 ± 0.05 ^b^	3.29 ± 0.2 ^b^	105 ± 8 ^c^	86 ± 3 ^c^
PB50	76.8 ± 0.8 ^ab^	33.1 ± 0.1 ^a^	0.89 ± 0.01 ^a^	324 ± 4 ^a^	3.18 ± 0.13 ^a^	34.1 ± 0.5 ^a^	3.13 ± 0.21 ^a^	2.78 ± 0.1 ^cd^	136 ± 6 ^a^	127 ± 3 ^a^
Cws	69.1 ± 0.8 ^d^	26.1 ± 0.1 ^d^	0.75 ± 0.04 ^bc^	256 ± 5 ^d^	2.85 ± 0.06 ^bc^	27.3 ± 0.5 ^c^	2.49 ± 0.04 ^c^	4.20 ± 0.1 ^a^	96 ± 10 ^d^	78 ± 3 ^d^
Ci	78.8 ± 0.7 ^a^	24.1 ± 0.1 ^g^	0.71 ± 0.02 ^c^	231 ± 4 ^e^	2.69 ± 0.08 ^d^	25.8 ± 0.5 ^d^	2.27 ± 0.03 ^d^	2.52 ± 0.1 ^d^	86 ± 9 ^e^	67 ± 3 ^e^

**Table 2 plants-10-00387-t002:** Means ± SD (n = 4) of the ascorbate peroxidase (APX), catalase (CAT), and glutathione peroxidase (GPX) enzyme activities in rice leaves according to different treatments and after 10 days of drought stress. PB3, drought stress with *B. endophyticus* PB3 foliar spray; PB46, drought stress with *B. altitudinis* PB46 foliar spray; PB50, drought stress with *B. megaterium* PB50 foliar spray; Cws, drought stress control; and CI, irrigated control. Values with different letters (above the bars) are significantly different according to Duncan’s test (*p* ≤ 0.05).

Treatments	Enzyme Activities (mmole min^–1^ mg protein^–1^)		
	APX	CAT	GPX
PB3	3.46 ± 0.08 ^b^	66.61 ± 1.9 ^bc^	0.210 ± 0.04 ^b^
PB46	2.83 ± 0.06 ^c^	70.28 ± 3.7 ^b^	0.224 ± 0.04 ^b^
PB50	2.71 ± 0.10 ^c^	62.94 ± 3.4 ^c^	0.196 ± 0.02 ^b^
Cws	4.09 ± 0.07 ^a^	80.36 ± 4.3 ^a^	0.279 ± 0.03 ^a^
Ci	2.20 ± 0.08 ^d^	55.92 ± 5.4 ^d^	0.145 ± 0.03 ^c^

**Table 3 plants-10-00387-t003:** Primers details used in qPCR analysis.

Genes	NCBI Accession No.	Primer Sequence	Melting Temp. (°C)	Amplicon Length (bp)
*LEA*	XM_015782086	F—5′ GGATCACTAGACGCCGTGAA 3′ R—5′ CAGAAATCCTCCCCTGCGAC 3′	59.55 60.46	151
*HSP70*	XM_015776732	F—5′ GAATCGTGACGGTCTCAGCA 3′ R—5′ CGATGAGGGCTTTCCGTTCT 3′	60.11 60.11	156
*RAB16B*	XM_015762125	F—5′ ATCGATCGACGGCTTTGACA 3′ R—5′ GCCCCTGGTAGTTGTCCATC 3′	59.83 60.11	154
*bZIP23*	XM_015770367	F—5′ AGATCACGCTGGAGGAGTTC 3′ R—5′ CGGAGGGAACACATTGCTCT 3′	59.18 60.04	167
*AP2/* *ERF*	XM_015785947	F—5′ GTGACAGCACAGTCACAACG 3′ F—5′ GATGACGAGGCTACCTTCACC 3′	59.70 60.20	144
*SNAC1*	XM_015775072	F—5′ TGGACCTGAGCTACGACGAT 3′ R—5′ TCACCTCAGAACGGGACCAT 3′	60.39 60.54	157
*Actin*	XM_015761709	F—5′ GGACTCTGGTGATGGTGTCA 3′ R—5′ TTTCCCGTTCAGCAGTGGTA 3′	59.02 59.24	164

NCBI, National Centre for Biotechnology Information; bp, base pairs.

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
