# Peer review of "The Foliar Application of Rice Phyllosphere Bacteria induces Drought-Stress Tolerance in Oryza sativa (L.)"

_plants, 2021, doi:10.3390/plants10020387_

Round 1

Reviewer 1 Report

Dear Authors,

this manuscript presents an interesting work about the physiological and transcriptional effect of PGP bacteria application on rice under water stress conditions.

The methods are well explained, and results are well presented with a deep statical analysis.

The included figures are needed to understanding the manuscript and some of them are highly illustrative.

In my opinion, this work can be published in the present form.

Best regards

Author Response

Thank you for the comments.

Reviewer 2 Report

In the manuscript “ The foliar application of rice phyllosphere bacteria confers induced systemic tolerance against water stress in Oryza sativa (L.)”, Devarajan et al. reported the effects of foliar application of Bacillus strains on controlling rice drought tolerance. They examined the physical and biochemical responses of rice after foliar application of bacteria and identified that B. megaterium PB50 significantly reduced stress-induced physical and cellular damage. Overall, this manuscript demonstrates the great potential of PGPB in alleviating drought stress responses in rice. Here are some specific comments:

  1. The authors emphasized the systemic tolerance against water stress in the title. I think it is not exactly fit the content.
    (1) All the responses that they examined are in the aboveground tissues. How to tell the response the systemic tolerance?
    (2) Generally, water stress includes drought and waterlogging. Because the authors only examined drought stress responses in this manuscript, I suggest to make the title more specific.
  2. Please make the content precise and concise. For example,
    (1) in line 184-186 “The results show relatively little variation in the expression of the genes between treatment replicates. The relative upregulation differed between genes and was also significantly different between treatments.” It is easy to tell the variance between biological replicates from the size of error bars and the significance of gene expression between treatments from the letters above bars. The authors only need to specify the number of biological replicates and the error bars derived from SD or SE in the figure and describe the change of gene expression level after treatments.
    (2) in line 355 “NAC transcription factors are encoded by NAC ” I think most readers know that NAC transcription factors are encoded by NAC. It is not necessary to emphasize the fact.
    (3) in line 360-363 “Conversely, PB50 could induce SNAC1 expression through an ABA-independent path way via acetoin since this strain can efficiently produce 3-hydroxybutanone (acetoin)…” It is not easy to understand the involvement of acetoin in the induction of SNAC1 and why the authors need to emphasize the biotransformation of acetoin to 2,3-butanediol. Please provide more detail and discuss the possible way of SNAC induction in a more logical way.
  3. In section 2.5, first, the authors showed the effects of applying megaterium PB50 on stomata closure, but the title of this section is the association of PB50 strain on rice leaf surface. The data shown in the beginning is not fit to the title. I suggest either to change the title or to move the data to section 2.1.
  4. The order of table 1 and table 2 is wrong. And the subtitles of section2 are not arranged well.
  5. The following are the comments about material and methods
    (1) in line 411-412 “All treatments and controls were performed in four replicates.” Please specify the number of plants per treatment in a single replicate.
    (2) in line 431 “A low-volume sprayer was used to spray 15–25 mL of inoculant per plant.” Please provide the detail about the frequency of foliar application of inoculum.
    (3) Figure 9 shows the difference of shoot morphology between treatments. I suggest to move the figure to the result section.

Reviewer 3 Report

the experiments were well conducted with clear and simple description of the material and methods utilized
The introduction of this paper gives an interesting description of the current study and at the end of section reports a clear purpose of work.
Results and Discussion:
Results are clearly exposed and this section presents good references, even very recently, and the authors have well compared the results obtained.
The tables and figures are very clear.

Author Response

Thanks for the comments.